# Can Extensive Training Transform a Mouse into a Guinea Pig? An Evaluation Based on the Discriminative Abilities of Inferior Colliculus Neurons

**DOI:** 10.3390/biology13020092

**Published:** 2024-02-02

**Authors:** Alexandra Martin, Samira Souffi, Chloé Huetz, Jean-Marc Edeline

**Affiliations:** Paris-Saclay Institute of Neuroscience (Neuro-PSI, UMR 9197), CNRS & Université Paris-Saclay, 91400 Saclay, France; alexandra.martin2@universite-paris-saclay.fr (A.M.); samira.souffi@mail.huji.ac.il (S.S.); chloe.huetz@universite-paris-saclay.fr (C.H.)

**Keywords:** inferior colliculus, neuronal recordings, mice, behavioral discrimination, masking noise

## Abstract

**Simple Summary:**

Is our ability to discriminate speech in noise an innate and fixed ability, or can it be increased by a perceptual learning task? And if this is the case, can we find neural correlates of this improved perceptual ability? To address this question, we trained adult mice to discriminate between acoustically similar calls from guinea pigs, first in quiet, then in two types of masking noise at three different signal-to-noise ratios. The mice performance was excellent in quiet and decreased when the signal-to-noise became negative. After three months of training, we recorded the auditory responses in auditory brainstem neurons (inferior colliculus) and found that their responses tended to be stronger and more discriminative than those of passively exposed mice and, more surprisingly, sometimes stronger and more discriminative than those of guinea pigs. These results exemplify the fact that extensive practice in a perceptual learning task can improve the way auditory neurons process speech-like sounds in noise.

**Abstract:**

Humans and animals maintain accurate discrimination between communication sounds in the presence of loud sources of background noise. In previous studies performed in anesthetized guinea pigs, we showed that, in the auditory pathway, the highest discriminative abilities between conspecific vocalizations were found in the inferior colliculus. Here, we trained CBA/J mice in a Go/No-Go task to discriminate between two similar guinea pig whistles, first in quiet conditions, then in two types of noise, a stationary noise and a chorus noise at three SNRs. Control mice were passively exposed to the same number of whistles as trained mice. After three months of extensive training, inferior colliculus (IC) neurons were recorded under anesthesia and the responses were quantified as in our previous studies. In quiet, the mean values of the firing rate, the temporal reliability and mutual information obtained from trained mice were higher than from the exposed mice and the guinea pigs. In stationary and chorus noise, there were only a few differences between the trained mice and the guinea pigs; and the lowest mean values of the parameters were found in the exposed mice. These results suggest that behavioral training can trigger plasticity in IC that allows mice neurons to reach guinea pig-like discrimination abilities.

## 1. Introduction

In natural conditions, human speech and animal communication sounds almost systematically occur with many other competing acoustic signals. Over the last two decades, a large number of studies have described the impressive resistance of auditory cortex (ACx) responses to communication sounds when the latter are presented in situations of acoustic degradations, especially in different types of masking noises [1,2,3,4,5,6]. Furthermore, in several studies performed either in birds or in rodents, it was claimed that the neural discrimination based on the spike timing of cortical neurons was correlated with the behavioral discrimination performance [2,3,7]. Therefore, these studies have prompted the view that perceptual robustness mainly relies on the capacity of cortical neurons to extract invariant acoustic features (e.g., [6]).

However, recent studies have challenged this view by comparing the discriminative abilities of auditory neurons at five levels of the auditory system: by quantifying the neuronal discrimination between acoustically similar stimuli (four guinea pig whistles) and using Mutual Information (MI) as a metric, it was reported that thalamic and brainstem neurons largely outperformed cortical neurons [8,9]. More precisely, both the individual (from individual recordings, MI_ind_) and the populational discriminative abilities (from small populations of recordings, MI_pop_) were higher in the auditory thalamus, inferior colliculus and cochlear nucleus than in the auditory cortex, both in silence and noisy conditions. Surprisingly, in the more challenging conditions, i.e., in a −10 dB signal-to-noise ratio (SNR), inferior colliculus (IC) neurons showed the highest discrimination abilities, suggesting that the neurons of this brainstem structure are potentially responsible for robust representations of target stimuli in noisy conditions. One potential explanation is that the envelope tracking abilities of neurons are higher in the brainstem than in the cortex, and these abilities are preserved in noisy conditions [9].

An intriguing question rarely addressed by previous studies is whether the quality of neuronal discrimination between heterospecific vocalizations can be enhanced by extensive behavioral training (but see [10] for this type of study). If this is the case, to what extent can the discrimination performance induced by such training be superior to the one observed in a species which produces these vocalizations and relies on them for their behavior in their daily life?

In the present study, we evaluated to what extent the inferior colliculus neurons of mice submitted to an extensive behavioral training between two guinea pig vocalizations belonging to the same category of whistle calls can display higher discrimination abilities than passively exposed mice, and, if so, are these abilities higher than those observed in the inferior colliculus of guinea pigs which use these vocalizations in their daily social interactions?

## 2. Materials and Methods

The experiments were performed under the national license A-91-557 (APAFIS project 40238) using procedure No. 32-2011 validated by the Ethics Committee in Animal Experimentation (CEEA 59 Paris Centre et Sud). All procedures were performed in accordance with the guidelines established by the European Communities Council Directive (2010/63/EU Council Directive Decree).

### 2.1. Subjects

Seven-week-old CBA/J female mice (n = 20) were obtained from Charles River (L’Arbresle, France). They were housed in a humidity (50–55%) and temperature (22–24 °C)-controlled facility on a 12 h/12 h light/dark cycle (light on at 7:30 A.M.) with free access to food. After one week of familiarization with the animal facility, all mice underwent a brief (30 min) surgery under isoflurane anesthesia (1.5–2%) to secure a small metallic hook (0.25 g) to the skull bone with super-bond C&B (Sun Medical, Dreux France) and dental acrylic cement (Methax, Makeval LTD, Essex UK). Metacam (0.1 mg/kg, i.p.) was administered after the surgery to help the animal recovery. Each mouse was weighed daily, and all mice included in this study re-gained weight after this initial surgery.

Two groups of mice were established: a group of trained mice (n = 10) and a group of exposed mice (n = 10). We randomly formed pairs of trained and exposed mice. During the behavioral training, each trained mouse had its own exposed mouse located next to it (less than 30 cm away), which was submitted to exactly the same number of stimuli and the same duration of restrained head-fixed condition in the acoustic chamber (IAC, model AC1). Both the exposed and the trained mice were under water restriction during the night. The exposed mice had access to water during the day before and after being in restrained head-fixed condition for 30–45 min; the trained mice had access to water during the behavioral training. Additional amounts of water were provided to the trained mice at the end of the day to maintain their body weight.

Electrophysiological data were also obtained from anesthetized guinea pigs (n = 10). Adult pigmented guinea pigs (aged 3 to 6 months, weighting from 606 to 1033 g, mean 812 g) came from our own colony and were housed in a humidity (50–55%) and temperature (22–24 °C)-controlled facility on a 12 h/12 h light/dark cycle (light on at 7:30 A.M.) with free access to food and water.

### 2.2. Behavioral Training

Mice were trained to an auditory discrimination task with a training procedure similar to those described in previous studies [9,11,12,13]. First, each mouse was habituated to being restrained by progressively increasing the time during which the mouse was head-fixed with the hook screwed to a small metallic post (from 10 min to 40 min over a week) in a red plastic tube and rested on aluminum foil. Mice could freely move their paws and body in the plastic tube but their head remained in a fixed position. Next, mice had 2–5 habituation sessions to learn to obtain a water reward by licking on a stainless steel water spout at least 8 times during the second after onset of the positive stimulus S+. A trial only started when the mice were not licking the spout for at least 3 s. Licks were detected by changes in resistance between the aluminum foil and the water spout. At the end of the habituation phase, the fraction of collected rewards was ~90%. Mice were then water-deprived and trained daily for about 150 trials in a Go/No-Go task involving two guinea pig whistles, one (the S+) signaling the reward (a 5 µL drop of water) and the other not (the S−, no water reward). Licking at the S− presentation delayed the following trial by 5 s.

After 5 sessions of conditioning to the S+ only, the discrimination protocol started in which mice received the S− for which they had to lick fewer than 8 times to avoid the 5 s time-out. One stimulus (the S+ or the S−) was presented every 10–20 s (uniform distribution) followed by a 1 s test period during which the mouse had to lick at least 8 times at the S+ to receive the reward. Positive and negative stimuli were played in a pseudorandom order with the constraint that 3 positive and 3 negative sounds must be played every 6 trials. The S+ and S− were guinea pig whistles used in several previous electrophysiological studies to assess the neuronal discrimination performance (indexed by mutual information) in the auditory system, from cochlear nucleus to auditory cortex [8,9,14]. The two whistles mostly differ in their temporal envelopes but also have slightly different spectral content (see Figure 1A in [8,9]). Once a mouse showed at least 80% correct discrimination between the S+ and the S− (computed as the average of correct responses to the S+ and the S−) and for two successive days in quiet conditions, it was trained in noisy conditions, first with the stationary noise (successively at +10, 0 and −10 dB SNRs), then with the chorus noise (successively at +10, 0 and −10 dB SNRs). To generate the chorus noise, audio recordings were performed in the colony room where a large group of guinea pigs were housed (30–40; 2–4 animals/cage). Several 4 s audio recordings were added up to generate a “chorus noise”, the power spectrum for which was computed using the Fourier transform. This spectrum was then used to shape the spectrum of a white Gaussian noise. The resulting vocalization-shaped stationary noise therefore matched the chorus noise audio spectrum, which explains why some frequency bands were overrepresented in the stationary noise (see Figure 1(A2,B2)).

In both the stationary and chorus noise, each mouse had to perform on two days at least at 80% in a given SNR to be tested on the following day at a lower SNR.

### 2.3. Recording Procedures

After 3–4 months of behavioral training and 3–5 days after the last behavioral session, the mice were anesthetized by a mixture of Ketamine and Xylazine (95 mg/kg ketamine, 24 mg/kg xylazine, i.p.) and a craniotomy was performed above the inferior colliculus. A multi-electrode array was lowered and multi-unit recordings were collected in one or two positions. Extracellular recordings were obtained using 16-channel multi-electrode arrays (ATLAS probes, ATLAS Neuroengineering, Leuven, Belgium) composed of one shank (10 mm) of 16 electrodes spaced by 110 µm. A stainless steel wire, used as ground, was connected to a screw inserted in the parietal bone. The raw signal was amplified 10,000 times (TDT Medusa, Alachua, USA). It was then processed by an RX5 multichannel data acquisition system (TDT). The signal collected from each electrode (sampling rate 25 kHz on each channel) was filtered (610–10,000 Hz) to extract multi-unit activity (MUA). The trigger level was set for each electrode to select the largest action potentials from the signal with 1 ms precision. On-line and off-line examination of the waveforms suggests that the MUA collected here was made of action potentials generated by a few neurons at the vicinity of the electrode. As we did not use tetrodes, the results of several clustering algorithms [15,16,17] based on spike waveform analyses were not reliable enough to isolate single units with good confidence.

Exactly the same recording procedure was used to collect neuronal recordings from the IC of anesthetized guinea pigs (n = 10), which were part of another study [8]. These animals were born in our own colony (45–60 animals) and daily made extensive use of their vocalizations during their social interactions. Data from the two species were collected with the same protocol and analyzed with the same Matlab scripts.

### 2.4. Experimental Protocol during Neuronal Recordings

As inserting an electrode array into a brain structure unavoidably induces a deformation of this structure, a 15 min recovery time was allowed for the tissue to return to its initial shape, then the array was slowly lowered. Time–frequency response profiles (TFRPs) were used to assess the quality of our recordings and to adjust electrode depth. When a clear frequency tuning was obtained for at least 10 of the 16 electrodes, the stability of the tuning was assessed: we required that the recorded neurons displayed at least three (each lasting 6 min) successive similar TFRPs (i.e., with similar best frequencies) before starting the protocol. When the stability was satisfactory, the protocol was started by presenting the acoustic stimuli in the following order: we first presented the four whistles used in previous studies [8,9] at 75 dB SPL in their original versions (i.e., in quiet), then the vocalizations were presented in stationary noise and finally, in chorus noise at 65, 75 and 85 dB SPL. Thus, the level of the original vocalizations was kept constant (75 dB SPL), and the noise level was increased (65, 75 and 85 dB SPL). In all cases, each vocalization was repeated 20 times and all the loudness levels were in RMS values. Presentation of this entire stimulus set lasted 45 min. The protocol was re-started after lowering the electrode by at least 300 μm.

### 2.5. Data Analysis

#### 2.5.1. The Quantification of Responses to Pure Tones

The TFRPs were obtained by constructing post-stimulus time histograms for each frequency with 1 ms time bins. The firing rate evoked by each frequency was quantified by summing all the action potentials from the tone onset up to 100 ms after this onset. Thus, TFRPs were matrices of 100 bins in abscissa (time) multiplied by 129 bins in ordinate (frequency). All TFRPs were smoothed with a uniform 5 × 5 bin window for visualization (not for the data analyses). For each TFRP, the best frequency (BF) was defined as the frequency at which the highest firing rate was recorded. Peaks of significant response were automatically identified using the following procedure: a positive peak in the TFRP was defined as a contour of firing rate above the average level of the baseline activity (100 ms of spontaneous activity taken before each tone onset) plus six times the standard deviation of the baseline activity. Recordings without significant a peak of responses or with inhibitory responses (decreases in firing rate 3 standard deviations below spontaneous activity) were excluded from the data analyses.

#### 2.5.2. The Quantification of Evoked Responses to the Four Vocalizations

The responses to vocalizations were quantified using two parameters: (i) the firing rate of the evoked response, which corresponds to the total number of action potentials occurring during the presentation of the stimulus minus spontaneous activity and divided by the stimulus duration; and (ii) the trial-to-trial temporal reliability coefficient (named CorrCoef as in previous studies [18,19]) which quantifies the trial-to-trial reliability of the responses over the 20 repetitions of the same stimulus. This index was computed for each vocalization: it corresponds to the normalized covariance between each pair of spike trains recorded at presentation of this vocalization and was calculated as follows:$$CorrCoef=\frac{1}{N(N-1)}\sum_{i=1}^{N-1}\;\sum_{j=i+1}^{N}\;\frac{\sigma x_{i}x_{j}}{\sigma x_{i}\sigma x_{j}}$$
where N is the number of trials and σx*_i_*x*_j_* is the normalized covariance at zero lag between spike trains x*_i_* and x*_j_* where i and j are the trial numbers. Spike trains x*_i_* and x*_j_* were previously convolved with a 10 msec width Gaussian window. Based upon computer simulations, we have previously shown that this CorrCoef index is not correlated with the neurons’ firing rates [18].

#### 2.5.3. The Quantification of Mutual Information from the Responses to Vocalizations

The method developed by Schnupp and colleagues (2006) [10] was used to quantify the amount of information contained in the responses to vocalizations obtained with natural and noisy stimuli. This method allows for quantifying how well the vocalization’s identity can be inferred from neuronal responses. Neuronal responses were represented using different time scales ranging from the duration of the whole response (total spike count) to a 1 ms precision (precise temporal patterns), which allows for analyzing how much the spike timing contributes to the information. As this method is exhaustively described in [10,18], we only present below the main principles.

The method relies on a pattern-recognition algorithm that is designed to “guess which stimulus evoked a particular response pattern” [10] by going through the following steps: from all the responses of a subcortical or cortical site to the different stimuli, a single response (test pattern) is extracted and represented as a PSTH with a given bin size. Then, a mean response pattern is computed from the remaining responses for each stimulus class. The test pattern is then assigned to the stimulus class of the closest mean response pattern. This operation is repeated for all the responses, generating a confusion matrix where each response is assigned to a given stimulus class. From this confusion matrix, the Mutual Information (MI) is given by Shannon’s formula:$$MI=\sum_{x,y}\;p(x,y)\times {log}_{2}\left(\frac{p(x,y)}{p(x)\times p(y)}\right)$$
where x and y are the rows and columns of the confusion matrix, or in other words, the values taken by the random variables “presented stimulus class” and “assigned stimulus class”.

In our case, we used responses to the four whistles and selected the first 280 ms of these responses to work on spike trains of exactly the same duration (the shortest whistle being 280 ms long). In a scenario where the responses do not carry information, the assignments of each response to a mean response pattern is equivalent to chance level (here, 0.25, because we used 4 different stimuli and each stimulus was presented the same number of times) and the MI would be close to zero. In the opposite case, when responses are very different between stimulus classes and very similar within a stimulus class, the confusion matrix would be diagonal and the mutual information would tend to log2(4) = 2 bits. This algorithm was applied with different bin sizes ranging from 1 to 280 ms (see Figure 2B in [8] for the evolution of MI with temporal precisions ranging from 1 to 40 ms). The value of 8 ms was selected for the data analysis because the MI reached its maximum at this value of temporal precision.

The MI estimates are subject to non-negligible positive sampling biases. Therefore, as in [10], we estimated the expected size of this bias by calculating MI values for “shuffled” data, in which the response patterns were randomly reassigned to stimulus classes. The shuffling was repeated 100 times, resulting in 100 MI estimates of the bias (MI_bias_). These MI_bias_ estimates are then used as estimators for the computation of the statistical significance of the MI estimate for the real (unshuffled) datasets: the real estimate is considered significant if its value is statistically different from the distribution of MI_bias_ shuffled estimates. Significant MI estimates were computed for MI calculated from neuronal responses under one electrode and for each condition. Therefore, there was a MI_bias_ value for each MI estimate.

**Figure 2 biology-13-00092-f002:**
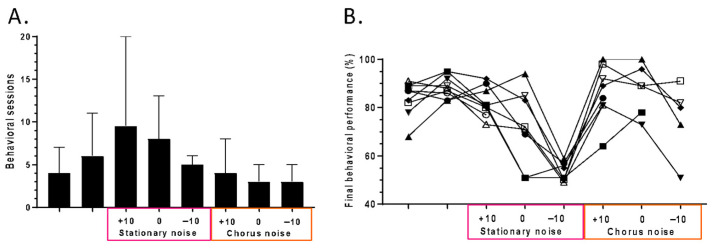
Behavioral results. (**A**) Median number of sessions before reaching the criterion of 80% correct responses on two successive days in the different stages of the protocol. Error bars represent the quartiles. Note that for the stationary noise at the −10 dB SNR, the animals stayed at 50% of correct responses (chance level) for six sessions and were stopped after these six sessions, as they showed no improvement. (**B**) Individual performances of the nine trained mice across the different stages of the protocol. The performances reported here are those obtained at the last session before reaching the criterion at each stage of the protocol. The performance dropped at the chance level in the −10 dB SNR for the stationary noise but recovered in the chorus noise.

### 2.6. Statistical Analysis

We used an analysis of variance (ANOVA) for multiple factors to reveal the main effects in the whole data set. Post hoc unpaired *t*-tests were performed between the different groups in quiet conditions and noisy conditions. All the analyses were performed on MATLAB 2021 (MathWorks).

## 3. Results

The behavioral results presented here were obtained from 10 extensively trained mice and 10 passively exposed mice. The electrophysiological results were obtained from nine trained mice, eight passively exposed mice and 10 guinea pigs. All the neuronal recordings included in the present study met two criteria: They showed significant time–frequency response profiles (TFRP) and responded significantly to at least one whistle in quiet conditions (i.e., the evoked firing rate was above spontaneous activity as assessed by a Wilcoxon rank-sum test).

Figure 1A shows the spectrograms and the envelopes of the two whistles used as S+ and S− in the original discrimination task (Figure 1(A1)), and it also displays the spectrograms of the S+ and S− signal in the stationary noise and in the chorus noise used in the following sessions (Figure 1(A2,A3), respectively). Figure 1B shows the spectrograms of the four whistles used to test the neuronal responses in quiet (Figure 1(B1)) and also displays the spectrograms of these whistles in the stationary and chorus noise (Figure 1(B2,B3), respectively). Figure 1C shows the timeline for the three groups of animals used in this study.

### 3.1. The Behavioral Performance of the Trained Mice in Quiet and in Noisy Conditions

The behavioral performance of the trained mice regularly progressed across the different stages of the protocol, and most of the mice mastered the increasing levels of difficulty in 3–10 sessions. Figure 2A shows the median number of sessions required to reach the criterion defined as 80% of correct responses to both the S+ and S− stimuli for two successive days. During the initial conditioning, five sessions were necessary for the mice to learn the licking behavior for the S+ whistle. On average, a little bit more than five sessions were required to achieve a good discrimination between the S+ and S−, then a long phase (almost 10 sessions) was required for achieving a good performance in the stationary noise at the +10 dB SNR and at the 0 dB SNR. At the −10 dB SNR, the protocol was stopped after a few sessions (six, usually) because of poor performance without any improvement across a number of days. In the chorus noise, the mice immediately recovered a good performance at the +10 dB SNR and maintained them in various SNRs. Figure 2B shows that the mice’s individual performances were high at the +10 dB SNR in the stationary noise, but that it started to decrease for five mice at the 0 dB SNR and was at the chance level for the −10 dB SNR. All but one mouse recovered good performance levels in the chorus noise at the +10 dB SNR, and for some mice, a relatively good performance was maintained up to the −10 dB SNR.

### 3.2. The Quantification of Neuronal Responses in Anesthetized Mice and Guinea Pigs in Quiet Conditions

In quiet conditions, the quantifications of the evoked firing rate, temporal reliability and mutual information are presented in Figure 3 (see Appendix A). The ANOVA revealed between-groups statistical differences for the evoked firing rate (*p* = 0.022), the CorrCoef index (*p* = 0.0009) and the mutual information (*p* = 0.006). Figure 3A shows that compared to the mean value of trained mice, the mean evoked firing rate of IC neurons was significantly lower in the exposed mice (unpaired *t*-test, *p* = 0.051) and in the guinea pigs (unpaired *t*-test, *p* = 0.018). The firing rate of IC neurons in the exposed mice and the guinea pigs were not statistically different (unpaired *t*-test, *p* = 0.929).

As illustrated in Figure 3B, the quantification of the trial-by-trial reliability assessed by the CorrCoef index revealed that compared to the trained mice, the temporal reliability was significantly lower in the exposed mice (unpaired *t*-test, *p* = 0.003) and in the guinea pigs (unpaired *t*-test, *p* = 0.0008). The firing rate of IC neurons in the exposed mice and the guinea pigs were not statistically different (unpaired *t*-test, *p* = 0.946).

As displayed in Figure 3C, the quantification of the neural discrimination assessed by the mutual information (MI) revealed that compared to the mean value of trained mice, the mean MI value was not significantly lower in the exposed mice (unpaired *t*-test, *p* = 0.22) but was significantly lower in the anesthetized guinea pigs (unpaired *t*-test, *p* = 0.0024). Also, the mean MI value of IC neurons in the exposed mice and the guinea pigs were not statistically different (unpaired *t*-test, *p* = 0.17).

To summarize, in silence conditions, the firing rate, temporal reliability and discrimination performance of IC neurons recorded from trained mice were all significantly higher compared to the guinea pig recordings. A similar trend was found in comparison to exposed mice (significant difference in temporal reliability only).

### 3.3. The Quantification of the Neuronal Responses in Stationary Noise

Figure 4 (see Appendix A) represents the mean values of the evoked firing rate, temporal reliability and mutual information obtained in three levels of signal-to-noise ratio in stationary noise.

For the evoked firing rate, the ANOVA revealed between-groups statistical differences for the +10 dB SNR (*p* = 0.016), the −10 dB SNR (*p* = 0.046), but not for the 0 dB SNR (*p* = 0.15). For the temporal reliability, the ANOVA revealed between-groups statistical differences for the three SNRs (*p* < 0.001 for the +10 dB SNR, the 0 dB SNR and the −10 dB SNR, respectively). For the mutual information, the ANOVA revealed between-groups statistical differences for the +10 dB SNR (*p* = 0.022), the −10 dB SNR (*p* = 0.007) but not for the 0 dB SNR (*p* = 0.39).

Figure 4A shows that the evoked firing rate of IC neurons was not significantly different in trained mice than in guinea pigs, whereas it was systematically lower in the exposed mice. The difference between exposed mice and guinea pigs was significant at the three SNR levels (unpaired *t*-test, *p* = 0.0007; *p* = 0.0242 and 0.0025 at the +10 dB, 0 dB and −10 dB SNRs, respectively), whereas the difference between the trained mice and the guinea pigs was not significant (unpaired *t*-test, *p* = 0.180; *p* = 0.281 and 0.92 at the +10 dB, 0 dB and −10 dB SNRs, respectively). The difference between trained and exposed mice was not significant at the +10 dB or 0 dB SNR (unpaired *t*-test, *p* = 0.180; *p* = 0.47), but a trend was present at the −10 dB SNR (*p* = 0.055).

Figure 4B shows that the trial-to-trial reliability (CorrCoef) of IC neurons was not often lower in trained mice than in guinea pigs, and it was even lower in the exposed mice. The difference between exposed mice and guinea pigs was significant at the three SNR levels (unpaired *t*-test, *p* = 0.0007; *p* = 0.0242 and 0.0025 at the +10 dB, 0 dB and −10 dB SNRs, respectively), whereas the difference between trained mice and guinea pigs was significant at the +10 dB and 0 dB SNRs (unpaired *t*-test, *p* < 0.0001; *p* = 0.0097), but not at the −10 dB SNR (*p* = 0.295). The difference between the trained and the exposed mice was not significant at the +10 dB and 0 dB SNRs (unpaired *t*-test, *p* = 0.180; *p* = 0.47), but was significant at the −10 dB SNRs (*p* = 0.022).

Figure 4C shows that the mean value of mutual information of IC neurons was not significantly different in trained mice than in guinea pigs, whereas it tended to be lower in the exposed mice. The difference between exposed mice and guinea pigs was significant at the +10 dB SNR (unpaired *t*-test, *p* = 0.006) but it was no different at 0 dB and −10 dB SNRs (*p* = 0.17 and *p* = 0.24, respectively). The difference between the trained mice and guinea pigs was not significant at the +10 dB and 0 dB SNRs (unpaired *t*-test, *p* = 0.280; *p* = 0.90, respectively), but was significant at the −10 dB SNR (*p* = 0.03).

To summarize, the data obtained in stationary noise pointed out that for the IC neurons recorded in the exposed mice, the firing rate, the temporal reliability and the discrimination abilities tend to be lower than in the trained mice and in the guinea pigs. There were much fewer significant differences between the mean values obtained from trained mice and guinea pigs.

### 3.4. The Quantification of the Neuronal Responses in Chorus Noise

Figure 5 (see Appendix A) represents the quantifications of the evoked firing rate, temporal reliability and mutual information obtained at three levels of signal to noise ratio in the chorus noise.

For the evoked firing rate, the ANOVA revealed between-groups statistical differences for the 0 dB SNR (*p* = 0.029), but not for the +10 dB SNR (*p* = 0.163) and for the −10 dB SNR (*p* = 0.058). For the temporal reliability, the ANOVA did not reveal between-groups statistical differences (*p* = 0.091, *p* = 0.226 and *p* = 0.098 for the +10 dB SNR, the 0 dB SNR and the −10 dB SNR, respectively). For the mutual information, the ANOVA revealed between-groups statistical differences for the three SNRs (*p* = 0.025 for the +10 dB SNR and *p* < 0.001 for the 0 dB and the −10 dB SNRs).

Figure 5A shows that the evoked firing rate of IC neurons tended to be higher in trained mice than in guinea pigs and lower than in exposed mice. The difference between trained mice and guinea pigs was significant at the 0 dB SNR (unpaired *t*-test, *p* = 0.0182), whereas the difference between the exposed mice and the guinea pigs was significant at the −10 dB SNR (unpaired *t*-test, *p* = 0.045). The other differences, especially those between trained and exposed mice, were not significant at the +10 dB and 0 dB SNRs (unpaired *t*-test, *p* = 0.123; *p* = 0.066), but there was a trend at the −10 dB SNR (*p* = 0.055).

Figure 5B shows that compared to the trained mice, the trial-to-trial reliability (CorrCoef) of IC neurons was not significantly different in exposed mice and in guinea pigs, whereas it tended to be slightly higher in the trained mice. The difference between trained mice and guinea pigs was only significant at the −10 dB SNR (unpaired *t*-test, *p* = 0.033), whereas the difference between the trained mice and the exposed mice was only significant at the +10 dB SNR (unpaired *t*-test, *p* = 0.047). The difference between the exposed mice and guinea pigs was not significant at the three SNRs (unpaired *t*-test, *p* = 0.068; *p* = 0.860 and *p* = 0.0571 at the +10 dB, 0 dB and −10 dB, respectively).

Figure 5C shows that the mean value of the mutual information of IC neurons was systematically lower in exposed mice than in trained mice, whereas it tended to be similar in trained mice and in guinea pigs. The difference between trained and exposed mice was significant at the +10 dB SNR (unpaired *t*-test, *p* = 0.0243) and at the −10 dB SNR (*p* = 0.043), but it was not at the 0 dB SNR (*p* = 0.087). The difference between the exposed mice and the guinea pigs was significant at the three SNRs (unpaired *t*-test, *p* = 0.007; *p* = 0.0007 and *p* < 0.0001, respectively). The mean value of mutual information was similar in trained mice and guinea pigs except at the −10 dB SNR where the guinea pig mean value was significantly higher (*p* = 0.007).

To summarize, the data obtained in chorus noise pointed out that for the IC neurons recorded in the exposed mice, the firing rate, the temporal reliability and the discrimination abilities are all lower than in the trained mice. The mean values obtained from the IC neurons of guinea pigs were often similar to the values obtained from trained mice.

## 4. Discussion

The results of the present study indicate that mice can learn to discriminate between acoustically similar heterospecific vocalizations belonging to the same category (guinea pig whistles) both in quiet conditions and in two types of noise at signal-to-noise ratios that were more and more challenging (from +10 dB up to −10 dB). After three months of extensive training, neuronal recordings from inferior colliculus neurons collected under general anesthesia showed that the firing rate, the temporal reliability and the neuronal discrimination abilities were, in most conditions, higher in trained mice than in exposed mice passively submitted to the same stimuli. Comparisons with recordings obtained from the inferior colliculus neurons of guinea pigs indicated that their evoked responses clearly differed from those of exposed mice, but were in the range of those of trained mice.

### 4.1. Behavioral Training with Conspecific or Heterospecific Communication Sounds

Obviously, one can question if relevant information can be derived from training animals of a given species to discriminate communication sounds from another species.

On one hand, it can be considered that the most important acoustic information is that coming from congeners. With this logic, in songbirds, behavioral discrimination has often been tested with conspecific songs. For example, in the Narayan et al. (2007) study [2], zebra finches had to discriminate between six target songs (10–20 s in duration) and had to peck on the left or on the right depending on the song to earn the food reward. After that, the birds were trained to detect these target songs embedded in three types of masking noises (including a broadband noise and a chorus noise) at SNRs ranging from −24 to +36 dB. The birds’ behavioral performances were at the chance level when negative SNRs were used, whereas they were above 88% of correct responses in silence. Similarly, in the Go–No-Go protocol used in [3], zebra finches had to discriminate between songs (several seconds in duration) embedded in a chorus noise with SNRs ranging from +15 to −15 dB. The high bird performance obtained in quiet decreased at the chance level only between −5 and −10 dB, but it should be noted that in these two studies long duration stimuli allowed the animals to detect multiple acoustic cues in the chorus noise to discriminate between the target stimuli. Thus, the marked decrease in the performance of our mice at the −10 dB SNR is in line with previous results.

To the best of our knowledge, there were very few attempts to train mice to discriminate between conspecific vocalizations, and never in noisy conditions. In a study performed in CBA/CaJ mice, the discrimination performance was assessed between vocalizations, which either contained harmonic or upsweeps [19]. It was shown that CBA mice have difficulties in discriminating between the different upsweep vocalizations as well as between the different harmonic vocalizations, but they easily discriminate the upsweeps from the harmonic vocalizations. In this mouse strain, the discrimination performance is inversely correlated with the spectrotemporal similarity [20]. These results clearly differ from those obtained in C57BL/6 mice, a mice strain known for its accelerated age-related hearing-loss. In a discrimination task, female C57BL/6 mice were found to be highly sensitive to disruptions of song temporal regularities, and preferentially approached playbacks of intact over rhythmically irregular versions of male songs [20]. In contrast, female behavior was invariant to manipulations affecting the songs’ sequential organization, or the spectrotemporal structure of individual syllables. In C57BL/6 mice, the temporal regularity seems to be the key acoustic cue extracted from complex vocal sequences during goal-directed behavior [20].

On the other hand, one can consider that discriminating the vocalizations of other species is crucial for survival. With this consideration in mind, Schnupp and colleagues (2006) [10] trained ferrets to discriminate between three twitter calls of marmoset (used as Go stimuli) and eight heterospecific calls from different species (used as No-Go stimuli). The ferrets performed at 80% correct or better after 25 days of training and routinely exceeded 90% correct after 2 months of training, a performance that they quickly recovered even after a long training holiday. Compared to this seminal study, it is important to point out that the task difficulty for our mice was much higher. Indeed, the two whistles that the mice had to discriminate slightly differed in terms of spectral content with more important differences in their temporal envelopes (see Figure 1). Thus, even in quiet, the discrimination can only rely on a small set of acoustic cues, which were progressively masked as the noise level increased (see discussion in [8,9]). Our task difficulty is potentially comparable to studies in which animals had to discriminate between English speech syllables (consonant–vowel–consonant) with durations in the range of our stimuli (200–600 ms). In fact, it was reported that rats were able to accurately discriminate consonant sounds even in the presence of background noise that was as loud as the speech target and that the discrimination abilities of auditory cortex (ACx) neurons correlated with the discrimination performance [7,21,22,23,24].

Lastly, despite the fact that the calls used here were from the same general vocalization category (i.e., guinea pig whistle calls), we do not know if at the end of the training, the trained mice were classifying the S+ signals (in silence and in noise) in a different category as the other whistles (the S− and the two other whistles not present during behavioral training). Alternatively, the animals can also categorize the S+ and S− in a category of meaningful signals (signaling water occurrence or its non-occurrence) and the other whistles in another category (of non-meaningful sounds). Answering this question may be possible by using “catch trials” where the W3 and W4 are presented during the behavioral task, with the risk of reducing the animals’ performances.

### 4.2. The Neuronal Consequences of Training with Heterospecific Vocalizations

In quiet, our results indicate that the firing rate, the temporal reliability and the discriminative abilities of IC neurons tended to be the highest in trained mice. Surprisingly, it was not the case in the two types of noises: Both in the stationary and in the chorus noise, the firing rate and the discriminative abilities of IC neurons did not often differ between trained mice and guinea pigs (except in the −10 dB SNR), and, in contrast, the IC neurons of exposed mice almost systematically exhibited the lowest values of firing rates, temporal reliability and mutual information. Globally, it seems that in noisy conditions, the behavioral task has improved the neural processing of the IC neurons of the trained mice compared to the exposed mice, making their responses almost as resistant to noise as those of guinea pigs IC neurons (even if in the noise, the decrease in MI seems to be larger for the trained mice than for the guinea pigs). Both guinea pigs and trained mice used the vocalizations for their behavior: the guinea pigs used the vocalizations from birth for their daily social interactions; the trained mice have learned to discriminate these vocalizations for obtaining water rewards for three months. Obviously, it is difficult to evaluate whether the behavioral discrimination performances of mice and guinea pigs are comparable, and if the classification into categories generated by these two species are similar. However, it is interesting to note that in the most difficult condition, i.e., the −10 dB SNR, opposite results were found between the stationary noise and chorus noise: in stationary noise, the MI values were larger in trained mice than in guinea pigs, whereas it was the reverse in the chorus noise. Mice IC neurons had faced the stationary noise during training but not the guinea pigs’ IC neurons. In contrast, the guinea pigs’ IC neurons had faced many types of chorus noise from birth, which was not the case for the trained mice.

The effects of behavioral training with natural stimuli on the discriminative abilities of auditory neurons have only been described at the cortical level. In their seminal study, Schnupp and colleagues (2006) [10] reported that after several months of training, two-thirds of ACx neurons displayed significant values of information about the stimuli identity, whereas this was the case for only one third of the neurons in naïve animals. In addition, many neurons in trained animals reached values >2 bits close to the maximal value (2.58 bits) that could be expected in their experiment [10]. Subsequent studies performed in rats ACx also confirmed that behavioral training can increase the neuronal discrimination performance in silence and in noise and that, after training, the spike-timing precision of ACx neurons correlated with the animal’s behavioral performance [7].

The present results bring two new findings: firstly, our data show that training with natural stimuli also modifies the discriminative abilities of auditory brainstem neurons. Secondly, the effects of behavioral training on the neural discriminative abilities are based upon a comparison between the responses of trained mice to those obtained in passively exposed mice which were submitted to exactly the same number of presentations of the S+ and S− stimuli. In a previous study performed in the gerbil ACx [25], it was reported that response variability decreased when the gerbils were engaged in detecting amplitude modulation noise compared to passively listening to the modulated noise. Note that a decrease in the response variability both in terms of firing rate and in terms of temporal structure may result in an increase in MI values, as shown in the present study.

Together, these results confirm that the consequences of active listening on auditory neurons differ from those induced by passive exposure. As explained below, the potential mechanisms involved in the effects triggered by active training may explain the differences with the passive exposure condition.

### 4.3. The Potential Mechanisms of This Neuronal Plasticity Occurring at the IC Level

Several decades ago, electrophysiological studies have reported reliable learning-induced plasticity occurring in the inferior colliculus [26,27,28] and even in the cochlear nucleus [29,30] after either appetitive or aversive conditioning. The results observed here after extensive training expanded these initial findings by showing that this plasticity can enhance the discriminative abilities of IC neurons compared to passively exposed animals. Whether this learning-induced plasticity is intrinsic to the IC, or transmitted from lower auditory structures (e.g., cochlear nucleus), and/or controlled by higher auditory structures (e.g., ACx or auditory thalamus) cannot be easily untangled except with targeted silencing ([31,32]; for review, see [33]). However, the fact that this plasticity can be observed under anesthesia suggests that the cortical influence is potentially limited as shown by studies where inactivating cortical neurons led to no effect on IC neuronal responses [34].

Over the last years, several studies have described the modulation exerted by Dopamine (DA) on the responses of IC neurons [35,36,37]. Nonetheless, the effects described in these studies are heterogeneous and cannot totally explain the present results. In fact, during the training situation, the reward was systematically associated with the S+ presentations, which generated a temporal pairing situation between the whistle used as S+ and the activation of the reward system. This situation should promote the release of DA but also of other neuromodulators such as Noradrenaline (NA) and Acetylcholine (ACh) in a close temporal relationship with the S+ occurrence. In the ACx, it has been documented that a temporal pairing between a particular sound frequency and the activation of the ventral tegmental area can increase the cortical responses and enlarge the neural representation of this frequency [38]. However, the long-lasting actions of NA and ACh (review in [39]) should trigger a long duration increase in neuronal excitability, which should persist during all the subsequent S+ and S− trials. At the synaptic level, it has long been known that DA modulates long-term potentiation and long-term depression [40,41]. By acting on the synaptic inputs reaching the IC neurons, DA can trigger plasticity for all types of whistles presented after the behavioral task, which all have a similar frequency range but different temporal envelopes. Thus, we can envision a scenario where the collective action of several neuromodulators can both increase the firing rate and the temporal reliability for a class of stimuli (the whistle-types of stimuli) and also increase the neuronal discriminative abilities to differentiate between these stimuli. Although speculative, this scenario can be submitted to several types of empirical tests. For example, we can apply low concentrations of DA antagonists in the IC to evaluate whether this will impair the behavioral and neuronal discrimination performance in our task. Also, it should be interesting to reveal changes in synaptic efficacy by testing the synaptic inputs converging onto collicular neurons. In vitro studies have shown that several types of IC neurons can display an NMDA-dependent form of long-term potentiation (LTP, e.g., see [42,43]) when stimulating the IC afferents with appropriate parameters. A large number of cellular mechanisms can be responsible for this LTP phenomenon in the IC such as increases in post-synaptic receptors’ density; increases in glutamate release or reduction in GABAergic release. It should be very valuable to compare the post-synaptic receptor density in trained and exposed mice after the completion of the behavioral task. Testing these different mechanisms can be a challenge in future research.

## 5. Conclusions

CBA mice can be trained to discriminate heterospecific vocalizations in quiet and in noise; some individuals can even perform well up to a 0 dB SNR. The responses of the inferior colliculus neurons of mice submitted to an extensive discrimination task between guinea pig vocalizations displayed more similarities with the responses of the inferior colliculus neurons of guinea pigs than with the responses of the inferior colliculus neurons of passively exposed mice. We conclude that a learned discrimination enhances both behavioral and neural performance up to a level similar to the innate one of another species.

## Figures and Tables

**Figure 1 biology-13-00092-f001:**
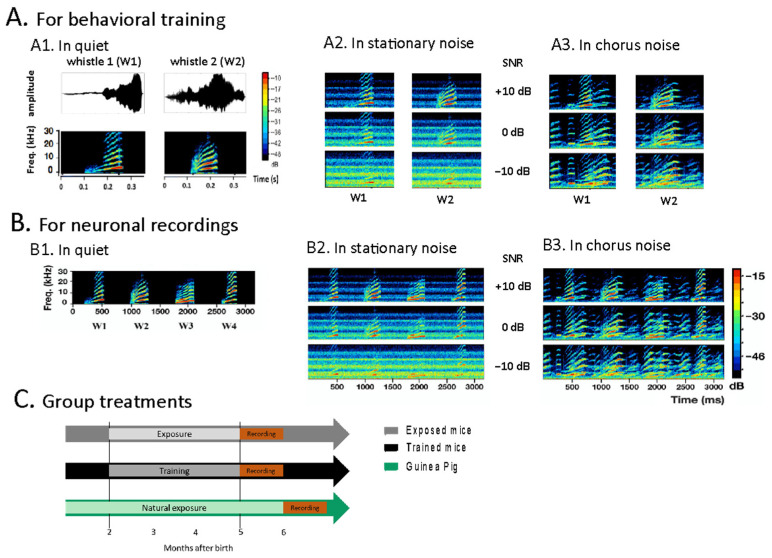
Stimuli and timeline of the experiment. (**A**) Stimuli used during the behavioral task. (**A1**) Waveforms (top) and spectrograms (bottom) of the two whistle stimuli used during the behavioral task. (**A2**) Spectrograms showing the whistles in stationary noise. (**A3**) Spectrograms showing the whistles in chorus noise. (**B**) Stimuli used for testing the neurons’ responses. (**B1**) Spectrograms of the four whistle stimuli used during the neuronal recordings. (**B2**) Spectrograms showing the four whistles in stationary noise. (**B3**) Spectrograms showing the four whistles in chorus noise. (**C**) Timeline of the experiment. Both the exposed and trained mice started the experiment at 2-months old. The trained mice were daily trained in the discrimination task first in quiet, then in the stationary noise, then in the chorus noise. The exposed mice were exposed in the same isolated booth as the trained mice and each trained mouse had its own yoked exposed mouse. The guinea pigs were naturally exposed to the whistle calls of the colony in their housing facility.

**Figure 3 biology-13-00092-f003:**
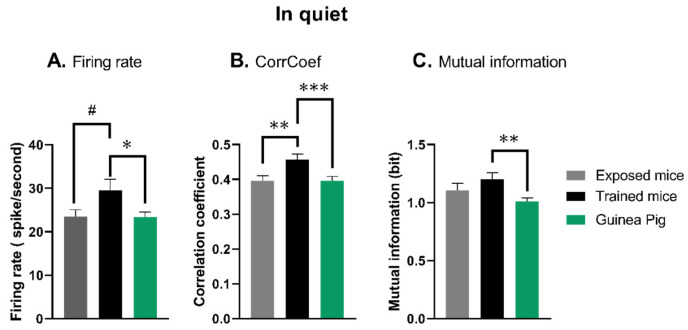
Quantifications of the neuronal responses of IC neurons in quiet. (**A**) Mean firing rate (±sem) obtained during the presentation of the four whistle stimuli for the neurons tested in exposed mice (grey bar), in the trained mice (black bar) and in the guinea pigs (green bar). (**B**) Mean CorrCoef (±sem) obtained during the presentation of the four whistle stimuli for the neurons tested in exposed mice (grey bar), in the trained mice (black bar) and in the guinea pigs (green bar). (**C**) Mean values of mutual information (±sem) obtained when analyzing the responses to the four whistle stimuli for the neurons tested in exposed mice (grey bar), in the trained mice (black bar) and in the guinea pigs (green bar). # *p* = 0.051; * = *p* < 0.05; ** = *p* < 0.01; *** = *p* < 0.001.

**Figure 4 biology-13-00092-f004:**
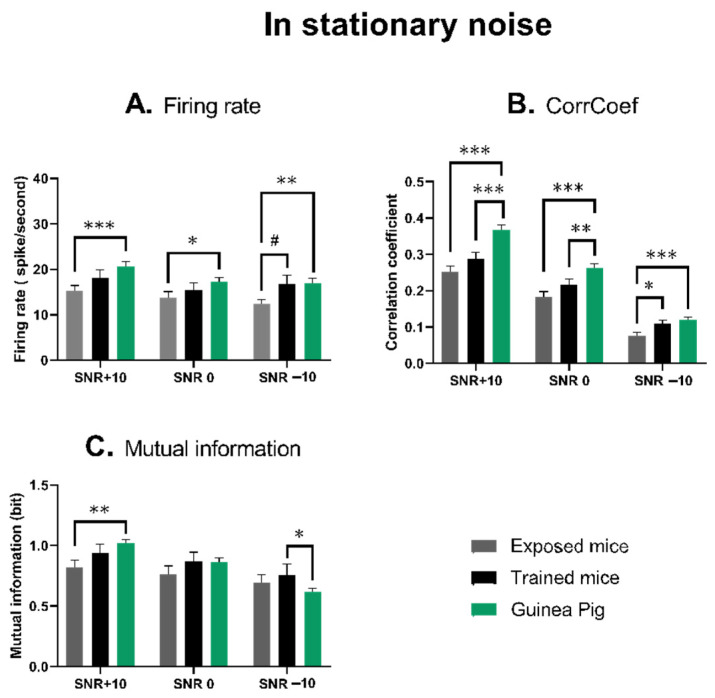
Quantifications of the neuronal responses of IC neurons in the stationary noise at the three SNRs. (**A**) Mean firing rate (±sem) obtained during the presentation of the four whistle stimuli for the neurons tested in exposed mice (grey bar), in the trained mice (black bar) and in the guinea pigs (green bar). (**B**) Mean CorrCoef (±sem) of neuronal responses obtained during the presentation of the four whistle stimuli for the neurons tested in exposed mice (grey bar), in the trained mice (black bar) and in the guinea pigs (green bar). (**C**) Mean values of mutual information (±sem) obtained when analyzing the responses to the four whistle stimuli for the neurons tested in exposed mice (grey bar), in the trained mice (black bar) and in the guinea pigs (green bar). # *p* = 0.055; * = *p* < 0.05; ** = *p* < 0.01; *** = *p* < 0.001.

**Figure 5 biology-13-00092-f005:**
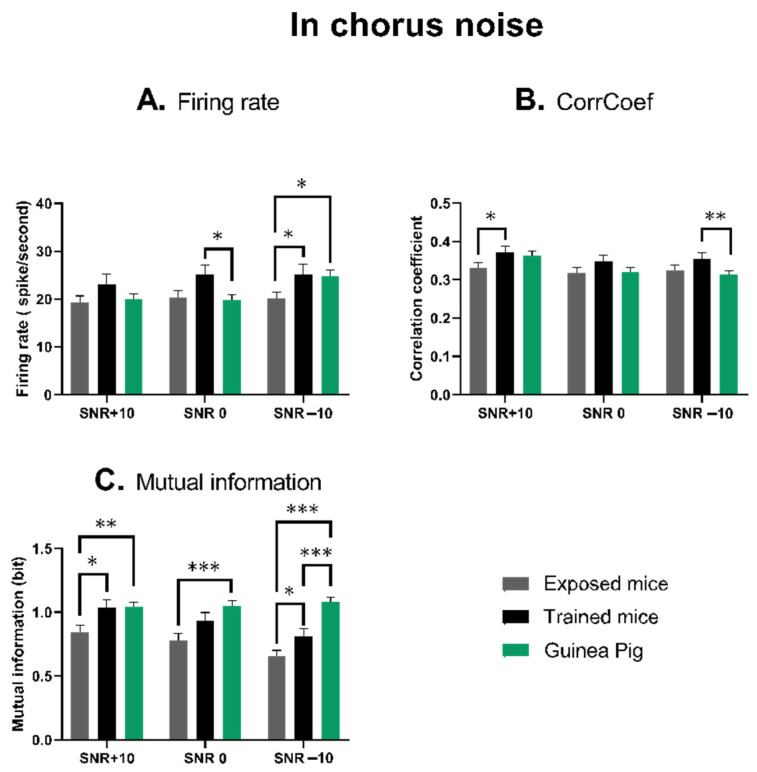
Quantifications of the neuronal responses of IC neurons in the chorus noise at the three SNRs. (**A**) Mean firing rate (±sem) obtained during the presentation of the four whistle stimuli for the neurons tested in exposed mice (grey bar), in the trained mice (black bar) and in the guinea pigs (green bar). (**B**) Mean CorrCoef (±sem) obtained during the presentation of the four whistle stimuli for the neurons tested in exposed mice (grey bar), in the trained mice (black bar) and in the guinea pigs (green bar). (**C**) Mean values of mutual information (±sem) obtained when analyzing the responses to the four whistle stimuli for the neurons tested in exposed mice (grey bar), in the trained mice (black bar) and in the guinea pigs (green bar). * = *p* < 0.05; ** = *p* < 0.01; *** = *p* < 0.001.

## Data Availability

The data that support the findings of this study are available from the corresponding author upon request.

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
