# Peer review of "Can Extensive Training Transform a Mouse into a Guinea Pig? An Evaluation Based on the Discriminative Abilities of Inferior Colliculus Neurons"

_biology, 2024, doi:10.3390/biology13020092_

Round 1

Reviewer 1 Report

Comments and Suggestions for Authors

In the present study, the authors trained mice to discriminate guinea pig calls in different contexts, silence and noisy conditions (stationary or chorus noise) and then compared the neural evoked activity to that of untrained but exposed to calls mice and guinea pigs. They conclude that training changes the neural responses of mice IC units with trained mice exhibiting greater firing rates, reliability across presentation and increased mutual information than untrained ones regardless of context. However, the differences between trained mice and guinea pigs were not so consistent across contexts, whereas in silence trained mice showed increased firing rate, reliability and MI that was not always the case in noise presentations although the responses from trained mice always resembled closer the ones from guinea pigs than from untrained mice did.

The study is well designed and the analysis is correct, the authors have a great deal of expertise on conspecific calls in Guinea pigs and on the effects of neuromodulation.

Mayor comment

The authors discuss their neural recording results in terms of neural conspecific or allospecific calls, assuming all calls form a unique conceptual category, rather than assuming different calls can belong to different categories based on their behavioural meaning. Only at the end of the discussion they touched on this idea when discussing possible mechanisms and the putative role of the Dopamine in establishing the reward association and long lasting effects of ACh and NA.

Whereas for guinea pigs different calls must have different behavioural meanings that could be context-dependent (as the authors indicate they are used in their social interactions). I do not think that is the case for trained mice that were taught by reward association to discriminate two particular calls whose meanings are constant. In addition, it is impossible to know what strategy the mice were using and if that strategy was constant across all the contexts it could be that some of the behavioural performance differences could be attributed to it.

 In silence, it could be a true discrimination between two, paying attention to both, or just a recognition of one therefore paying attention to only one and ignoring anything but the C+.  Therefore, those strategies could have an impact on the interpretation of the neural recordings and if the treatment of all calls as a single category is the best approach or if on the other hand neural responses to C+ or even C+ and C- should be considered another category than W3 and W4. It would be impossible to know what the trained mice extrapolate from their training and exposure to two calls in terms of perceptual categorisation.  This could partially explain the differences between in silence and in noise.

I think the discussion should be expanded to include the possible effect of categorisation.

Minor points

It is my understanding that W3 and W4 were not presented to the mice during behaviour or passive listening, but they constitute the 50% of the presentations during the recordings.

In Figure 2 if animals do not reach the criterion at stationary noise -10 and the animals are stopped at 6 sessions, the bar on panel A for that condition should be removed as it simply reflects the time of suspending the training.

In the methods, it says the animals are head-fixed. Does it mean they have undergone a previous surgery to implant a headpost to that end?

The animals that are not trained but are water regulated how do receive their water in an equivalent period of time as the trained ones? How much water do any of the mice receive during water regulation? It seems 33ml/g/day is too much water, I guess there is a mistake on the units ml/kg/day?.

Do the training animals receive only 0.7ml/session of water (5 microlitres x 150 trials)? Is there a maximum weight loss that triggers a welfare warning and supplementary water is given?

How long has it been since finishing the behavioural testing to the actual recording session?

Although all recordings are under anaesthesia and therefore attention is not a consideration could be that the differences observed between in silence and in noise recordings for the trained mice could be due to adaptation caused by the categorisation induced by the training? In untrained mice the firing rate is less by adaptation because of repetition and in guinea pig because familiarity, and therefore lack of novelty, whereas in trained mouse the representation of calls are enhanced?

It would be good to see if for the trained mice there is a correlation between the behavioural performance of the animals and the MI derived from the recordings.

Author Response

We would like to thank the reviewer for the comments that have contributed to improve our article. We have made changes which at detailed in the attached files

Reviewer 2 Report

Comments and Suggestions for Authors

This manuscript describes an elegant study, where 10 mice were trained in an auditory discrimination task involving two guinea pig vocalizations, and 10 other mice was passively exposed to exactly the same stimuli. After training and exposure, neural responses were measured from the inferior colliculus in the trained and exposed animals, as well as in 10 guinea pigs, who presumably had been exposed to (and had generated) similar vocalizations naturally. The results are ambiguous, and the authors claim more than their data will support. However, I believe these issues can be fixed with a revision.

The main problem is that the results are not as clear cut as the authors would like to assert. The authors claim in several places in the manuscript that the neural discrimination of the trained mice was better than that of the exposed mice. However, based on the statistical analysis, that was true even at the 0.05 level in 5 out of the 21 conditions tested. One of those 5 conditions was at -10 dB SNR in stationary noise, where behavior was at chance, and the stimuli were likely inaudible. For this reason, the authors’ main conclusion is not very well supported. Therefore, throughout the manuscript, any conclusions along those lines need to be severely toned down. While it is true that there is a trend for the trained mice’s responses to be higher than the exposed mice, the story is not clear cut. I would certainly dispute the assertion that the trained mice look more like guinea pigs. In fact, in quiet the guinea pig responses look more like those of the exposed mice. The readers would be better served with a more measured and accurate description of the data, based on the statistical outcomes.

Another issue relates to the measures used, namely mean firing rate, CorrCoef, and mutual information. I’m not sure what the mean rate signifies: Is it just the mean rate pooled across the two stimuli (S+ and S-)? Wouldn’t a more meaningful hypothesis be the difference in rate between S+ and S- if the mice are being trained to discriminate between them? I suppose that is partly what is derived via mutual information, but a clearer rationale would be helpful. The authors should also refer to relevant cortical work, showing that training did not result in changes in mean firing rate, but instead a change in the variance of the firing, so that responses were less variable, and hence more discriminable after training (e.g., von Trapp et al. J Neurosci, 2016, https://doi.org/10.1523/jneurosci.1302-16.2016).

Also, there was some confusion as to the stimuli themselves. I understood there to be two stimuli, but there are several references throughout the paper to four whistles, not two. This needs to be clarified.

Specific comments

Fig. 2. The bar graph is a little misleading, because it seems as if training only took 5-6 sessions to reach criterion at -10 dB, whereas performance was actually at chance and hence abandoned.

Statistics. Provide full details (t values, degrees of freedom, effect size) for all statistics.

Author Response

We want to thank the reviewer for these comments that have contributed to improve our article. The attached document provides all the answers to the reviewer's comment

Reviewer 3 Report

Comments and Suggestions for Authors

The manuscript describes behavioral and electrophysiological measurements in mice trained to discriminate two spectrally-distinct guinea pig vocalizations.  Control mice heard the vocalizations but were not trained.  Multineuron recordings from the inferior colliculus showed that training resulted in plastic changes in response measures that could be compared to previous observations in guinea pigs.  The experiments address interesting and important issues in auditory neuroscience.  Appropriate, technically-sophisticated methods were used for both types of experiments.  The Introduction and Discussion sections provide a good overview of this area of research.  

Specific comments

line 55.  I'm not sure I understand the meaning of the word "encompassed" as it is used here.  Does it indicate something like "outperformed"?

line 102.  The meaning of "head-fixed" isn't completely clear.  Presumably it indicates some degree of restraint, but how much?  A bit more clarification would help.

line 125:  A brief description of the chorus noise would be helpful.

line 504 (and previously at lines 121-122):  The claim that the two calls "mainly differed by their temporal envelope" is not convincing.  Fig 1 spectrograms show obvious differences in the 20-30 kHz region where mice are most sensitive.  If the authors think this is an important point to make (it doesn't seem important to me), they should provide a stronger justification.

Author Response

We want to thank the reviewer for all the relevant comments and hope that we have provided adequat answers in the attached document.

Round 2

Reviewer 2 Report

Comments and Suggestions for Authors

I have looked over the authors' comments and revised version. I continue to have reservations about the interpretation of the data, but I believe that the manuscript can be published as is.